# Mexican Plants Involved in Glucose Homeostasis and Body Weight Control: Systematic Review

**DOI:** 10.3390/nu15092070

**Published:** 2023-04-25

**Authors:** Montserrat Torres-Vanda, Ruth Gutiérrez-Aguilar

**Affiliations:** 1Laboratorio de Investigación en Enfermedades Metabólicas: Obesidad y Diabetes, Hospital Infantil de México “Federico Gómez”, Mexico City 06720, Mexico; 2División de Investigación, Facultad de Medicina, Universidad Nacional Autónoma de México (UNAM), Mexico City 04510, Mexico

**Keywords:** type 2 diabetes, glucose homeostasis, insulin secretion, body weight control, medicinal plants

## Abstract

Background: Obesity is defined as abnormal or excessive fat accumulation, provoking many different diseases, such as obesity and type 2 diabetes. Type 2 diabetes is a chronic-degenerative disease characterized by increased blood glucose levels. Obesity and type 2 diabetes are currently considered public health problems, and their prevalence has increased over the last few years. Because of the high cost involved in the treatment of both diseases, different alternatives have been sought. However, the general population uses medicinal plants, in the form of tea or infusions, to treat different diseases. Therefore, traditional medicine using medicinal plants has been investigated as a possible treatment for type 2 diabetes and body weight control. Aim of the study: The purpose of this review is to find medicinal plants used in Mexico that could exert their beneficial effect by regulating insulin secretion and body weight control. Material and method: For the development of this review, Mexican plants used in traditional medicine to treat type 2 diabetes and body weight control were searched in PubMed, Google Scholar, and Scopus. The inclusion criteria include plants that presented a significant reduction in blood glucose levels and/or an increase in insulin secretion. Results: We found 306 Mexican plants with hypoglycemic effects. However, plants that did not show evidence of an increase in insulin secretion were eliminated. Finally, only five plants were included in this review: *Momordica charantia* L. (*melón amargo*), *Cucurbita ficifolia bouché* (*chilacayote*), *Coriandrum sativum* L. (*cilantro*), *Persea americana* Mill. (*aguacate*) *Bidens pilosa* (*amor seco*), including 39 articles in total. Here, we summarized the plant extracts (aqueous and organic) that have previously been reported to present hypoglycemic effects, body weight control, increased secretion and sensitivity of insulin, improvement of pancreatic β cells, and glucose tolerance. Additionally, these effects may be due to different bioactive compounds present in the plants’ extracts. Conclusion: Both in vivo and in vitro studies are required to understand the mechanism of action of these plant extracts regarding insulin secretion to be used as a possible treatment for type 2 diabetes and body weight control in the future.

## 1. Introduction

Obesity is one of the most serious global public health problems, and it is defined as abnormal or excessive fat accumulation, representing a health risk. In adults, a body mass index (BMI) over 25 is considered overweight, and over 30 is obese [1]. The prevalence of obesity in children and adults has been increasing over the last decades, predicting that one billion people globally, including 1 in 5 women and 1 in 7 men, will be living with obesity by 2030 [2]. In Mexico, 76% of adult women are overweight or obese, while men have a prevalence of 72.1% [3]. Obesity increases the risk of developing different diseases, such as arterial hypertension, dyslipidemia, metabolic syndrome, and type 2 diabetes (T2D) [4].

T2D is a multifactorial metabolic disorder that is mainly influenced by the presence of obesity (80–90% of TD2 patients are overweight or obese), lack of physical activity, poor eating habits, and genetic factors [5]. T2D is characterized by the presence of chronic hyperglycemia, which appears when the body does not effectively use the insulin it produces or, ultimately, when the pancreas does not secrete enough insulin [6].

The prevalence of T2D has gradually been increasing, with a world prevalence of 9.3% [7]. The prevalence of T2D in Mexico has been increasing in the last eight years from 9.2% (2012) to 10.6% (2020) [3]. Therefore, T2D is considered one of the leading causes of death in Mexico, representing a mortality of 15.7% [8].

The high prevalence of obesity and T2D in the Mexican population brings serious consequences on the economy, both individually and collectively [9], since those patients require more frequent medical attention and a greater amount of medicines [10]. Therefore, the general Mexican population resorts to medicinal plants in the form of teas or infusions to ameliorate T2D and control body weight [11].

Medicinal plants have been used empirically (either as part of the diet, infusions, or extracts) to treat and improve T2D symptoms and body weight control. Despite being used empirically, the World Health Organization has guidelines on the safety monitoring of herbal medicines [12]. Different medicinal plants are now approved and recommended for medicinal use after they have been scientifically validated to ensure safety and efficacy, such as *Echinacea purpurea*, *Panax ginseg*, and *Passiflora incarnata*. However, the use of medicinal plants is not yet approved by the FDA [13].

The use of medicinal plants has been reported to have an effect on preventing and restoring pancreatic β cell damage caused by this disease. The positive effects of medicinal plants can be attributed to the content of their chemical compounds, such as flavonoids, polysaccharides, saponins, triterpenes, alkaloids, and phenolic compounds, that could exert their action on glucose homeostasis [14]. In addition, these medicinal plants can improve insulin secretion [14], which is necessary for the proper maintenance of glucose homeostasis.

Consequently, it is very important to study the ethnic plants that offer a beneficial effect on glucose metabolism, as well as their physiological effects. Therefore, these Mexican plants could be used as plausible new treatments for T2D and/or body weight control and could be accessible to the general population. Therefore, in this review, we will focus on describing medicinal plants that could have an action on the stimulation of insulin secretion, as well as body weight control.

## 2. Methods

### 2.1. Protocol and Registration

This systematic review was performed according to the Preferred Reporting Items for Systematic Reviews and Meta-Analyses (PRISMA).

### 2.2. Information Sources and Search Strategy

For the development of this review, Mexican plants used in traditional medicine to treat T2D with hypoglycemic properties, insulin secretion, and body weight control were searched in the databases PubMed, Google Scholar, and Scopus. The bibliographic search was performed using keywords or phrases such as: “antidiabetic Mexican plants”, “Mexican plants with hypoglycemic effect”, “Mexican plants and type 2 diabetes treatment” “antidiabetic Mexican plants”, “antidiabetic Mexican plants and insulin secretion”, “antidiabetic Mexican plants and beta cells”, “effect of Mexican medicinal plants in insulin secretion”, “Mexican plants with glucose lowering effect” and body weight control. Data from publications between 1999 and 2020 were included. The detailed search approach is described in Figure 1.

### 2.3. Eligibility Criteria

The main inclusion criteria for this review were studies that reported a decrease in glucose levels, an increase in insulin secretion, and body weight change using in vivo or in vitro models.

The exclusion criteria were studies suggesting a mechanism of action other than insulin secretion, studies referring to non-Mexican plants, and studies that reported plants that did not have an effect on glucose homeostasis (Figure 1).

### 2.4. Study Selection

In the screening or identification stage, all bibliographic material was retrieved by screening the titles and abstracts. The duplicated articles were removed. In the eligibility stage, the full texts of the publications were examined to assess the inclusion criteria. The agreement of two people was granted for the papers included in this systematic review.

## 3. Results

Our initial search identified 1114 publications in Pubmed, Google Scholar, and Scopus; however, 300 papers were duplicated, so they were discarded. Of the 814 non-duplicates, 306 Mexican medicinal plants were found for the treatment of T2D. However, 641 articles were also discarded because they fell within the exclusion criteria, leaving us with 173 articles that were reviewed in detail. Finally, a total of 39 articles were included in this systematic review of the following five plants: *Momordica charantia* L. (*Melón amargo*), *Cucurbita ficifolia bouché* (*chilacayote*), *Coriandrum sativum* L. (*cilantro*), *Persea americana* Mil. (*aguacate*), and *Bidens pilosa* (*amor seco*). In this review, studies in animal models, cell lines, and very few human trials were primarily included. However, other medicinal plants, such as *Psidium guajava*, *Opuntia ficus indica*, and *Aloe vera* have been tested in diabetic patients to evaluate their hypoglycemic effect reducing glucose levels in 10%, 9.2%, and 7.7%, respectively [15,16,17,18].

### 3.1. Momordica charantia L. (Melón amargo)

*Momordica charantia* L. is a medicinal plant cultivated in tropical and subtropical areas of Asia, South America, Africa, and the Caribbean. In Mexico, it is grown in Oaxaca, Quintana Roo, Chiapas, Tabasco, Veracruz, and Yucatán [19]. It belongs to the *Cucurbitaceae* family and is colloquially known as *Melón amargo* [20] (Figure 2).

*Momordica charantia* L. produces an edible fruit that is harvested for cooking. The seeds and skin are also edible [20].

In the articles reviewed for this plant, aqueous or organic extracts from leaves, pulp, or seeds were obtained to perform in vivo and in vitro studies to analyze its hypoglycemic effect.

The effect of the methanolic extract on the leaves of *Momordica charantia* L. was evaluated in a diabetic rat model (Table 1). It was observed that the extract presented hypoglycemic activity (50% glucose levels reduction) and a reduction in triglyceride levels at doses of 200 mg/kg and 400 mg/kg, comparable to glibenclamide. An improvement in the structure of pancreatic β-cells was shown. It was also observed that the diabetic group, treated with the extract, presented an elevation in body weight, reversing the body weight loss caused by diabetes [22].

Another study demonstrated the benefits of the aqueous extract of the leaves of *Momordica charantia* L. (doses 100 and 400 mg/kg), reducing glucose levels by 21%, as well as for triglycerides, HDL, LDL, creatinine, urea levels, and liver enzymes. On the other hand, the diabetic group presented a body loss of 25.11% compared to the normoglycemic group; however, when administering the extract, 14.33% of body weight regained was observed (Table 1) [23]. In another study, the aqueous extract of the whole fruit of *Momordica charantia* L. at a dose of 20 mg/kg in a neonatal diabetic rat model provoked 33.3% of hypoglycemic activity in 2–4 weeks after treatment. Moreover, an increase in serum insulin levels and decreased damage to pancreatic islets was observed [24].

In rabbits, the ethanol extract of the pulp of *Momordica charantia* L. at a dose of 1 mg/kg and 3 mg/kg provoked a decrease in blood glucose of 54.8% and 61.35%, and 13% and 17% increase in serum insulin, respectively. Regarding body weight, the doses presented a decrease of 1.19% and 37%, respectively. However, in normoglycemic control animals, this extract increased the body weight by 4.4% [25]. In C57BL/6 mice fed a high-fat diet, a similar extract was administered (ethanol extract of the pulp) at doses of 250 mg/kg and 500 mg/kg, observing a decrease in glucose blood levels (75%), an increase in insulin sensitivity and a decreased in insulin levels (30%), compared to the obese group. Moreover, the doses of 250 mg/kg and 500 mg/kg presented a decrease of 14 and 25% in body weight, respectively, compared to the mice fed with a high-fat diet. In this experiment, a decrease in fat mass and an increase in expression in SIRT1 (a protein involved in fatty acid beta-oxidation in the liver and insulin sensitivity) was observed (Table 1) [26]. On the other hand, proteins extracted from the pulp of *Momordica charantia* L., administered at doses of 1, 5, and 10 mg/kg in both normoglycemic and diabetic rats, showed an increase in plasma insulin concentrations and a reduction in plasma glucose, reaching a reduction of 43% with the dose of 10 mg/kg in diabetic rats. The effect of this protein extract at a dose of 10 μg/mL was investigated in myocytes (C2C12) and adipocytes (3T3L1), demonstrated an increase in glucose uptake of 28% and 35%, respectively, compared to their baseline control [27] (Table 1).

The hypoglycemic effect of saponins from the hydroalcoholic extract of the pulp of *Momordica charantia* L. was evaluated. The saponins were administered for 4 weeks, using doses of 400, 200, and 100 mg/kg (Table 1). This study showed that saponins could reduce glucose levels (10%) and insulin resistance in diabetic rats [28]. In addition, an improvement in lipid metabolism was demonstrated, preventing oxidative stress and regulating the insulin signaling pathway. In this experiment, the body weight was restored, and a decrease in fasting insulin levels was also obtained, as well as a protective effect of the β cells [28] (Table 1).

In another study, the effect of saponins (doses of 20, 40, and 80 mg/kg) as well as polysaccharides (doses of 500 mg/kg) extracted from the pulp of *Momordica charantia* L. with isopropyl alcohol in diabetic mice reduced glucose levels and insulin resistance (Table 1). In addition, the extracts increased p-AMPK (phosphorylated AMP-activated protein kinase) and improved antioxidant capacity, exhibiting a protective effect on pancreatic β cells. The dose of 80 mg/kg of saponins presented a better effect because it significantly reduced fasting blood glucose levels, improved insulin resistance, and restored body weight [29].

Other compounds found in *Momordica charantia* L. extracts are the triterpenes that are used at 20 and 30 μM in a mouse cell line of hepatocytes (FL83B), increased glucose uptake, and insulin receptor 1 (IRS-1) phosphorylation in insulin-resistant cells, which could indicate the improvement of insulin sensitivity. In addition, the triterpenes favored the translocation of GLUT-4 to the cell surface of insulin-resistant cells, increased AMPK activation and PTP-1B inhibition, improving the effects caused by insulin resistance [30].

In the articles reviewing this plant, the different extracts were compared primarily with glibenclamide (antidiabetic drug), having a similar hypoglycemic effect (50%) and restores body weight loss [22]. In addition, studies comparing the effect of these extracts against metformin (an antidiabetic drug) demonstrated a reduction of blood glucose levels of 50.1% using 300 mg/kg plant extract and 47% for metformin [31,32,33].

Therefore, *Momordica charantia* L. presents an effect in blood glucose and lipids (triglycerides, HDL, LDL) reduction, an increase in insulin sensitivity and secretion, an improvement in β cells, as well as restoring body weight. These effects can be due to the chemical compounds it presents (saponins, proteins, triterpenes, and polysaccharides) (Table 6).

### 3.2. Cucurbita ficifolia bouché (Chilacayote)

*Cucurbita ficifolia bouché* is a medicinal plant native to America. In Mexico, it is grown in Hidalgo, Guerrero, Michoacán, and Veracruz [34]. It belongs to the *Cucurbitaceae* family and is colloquially known as *chilacayote* [35] (Figure 3).

The skin, pulp, and seeds are edible. In Mexico, the pulp and seeds are widely used to prepare different dishes and regional sweets [35].

Various investigations have been carried out regarding its hypoglycemic effect, both in vivo and in vitro, mainly using the seeds and pulp of the plant. These studies focus on its effect on insulin secretion and the mechanism for lowering blood glucose levels (Table 2), which are described below.

A clinical study with 10 patients with T2D, to whom *Cucurbita ficifolia bouché* juice was administered at a dose of 4 mL/kg, showed a 31% decrease in glucose levels after 5 h of intake (Table 2); however, its effect on insulin levels was not evaluated [37]. Another study demonstrated the hypoglycemic effect of lyophilized *Cucurbita ficifolia bouché* fruit juice, both in normoglycemic and diabetic mice, administered intraperitoneally and orally (250, 500, 750, 1000, and 1250 mg/kg). A significant decrease in glucose levels (83%) was obtained in both normoglycemic and diabetic mice, but insulin levels were not measured (Table 2). However, its daily consumption in both cases caused acute toxicity, presenting less toxicity and greater hypoglycemic effects at 500 mg/kg [38].

Recently, *Cucurbita ficifolia bouché* seeds at different stages of ripeness in diabetic and normoglycemic mice at a dose of 500 mg/kg were tested (Table 2). A hypoglycemic effect was observed in all stages of ripeness, having the best effects at 15 days of maturity by reducing 50% of glucose levels. Furthermore, it is suggested that phenolic compounds may be responsible for their hypoglycemic effect, having some effect on insulin secretion [39]. The methanolic extract of *Cucurbita ficifolia bouché* seedless fruit administered in the diabetic group (300 and 600 mg/kg) and in normoglycemic rats (600 mg/kg) was studied. In both groups, a significant decrease in blood glucose levels was observed (60% in the diabetic group). Regarding the diabetic group, a decrease in glycated hemoglobin, body weight (25%), and an increase in plasma insulin (53%) were observed. The administration of 600 mg/kg extract restored the body weight by 6%; however, the dose of 300 mg/kg did not cause a difference in body weight. [40].

In another study, the methanolic extract of the seedless fruit was used in diabetic and normoglycemic rats (Table 2). The rats were fed orally with the extract at a dose of 300 mg/kg, showing a reduction in hyperglycemia (12.5%), pancreatic lipid peroxidation, and an increase in insulin levels in the diabetic group (36%), as well as an increase in the number of active pancreatic β cells [41].

The hypoglycemic effect of the aqueous extract of *Cucurbita ficifolia bouché* seedless pulp and D-quiro-inositol (a compound present in this plant) was evaluated using RINmF5 cells (mouse pancreatic β cells). It was shown that *Insulin* and *Kir6.2* (component of the potassium channel sensitive to ATP) gene expression levels increased in the cells (Table 2). This effect suggests a mechanism of action that involves the expression and secretion of insulin [42]. Based on the previous study, another in vitro investigation (RINmF5 cells) demonstrated that D-quiro-inositol and the extract did not increase insulin or calcium concentration (Table 2). However, the aqueous extract increased the concentration of calcium and insulin. Therefore, the hypoglycemic effect of this plant may be due to other chemical compounds [43], like phenolic compounds (gallic acid, chlorogenic acid), in addition to D-quiro-inositol (Table 6).

In the articles reviewed for this plant, the different extracts have been compared primarily with tolbutamide (83% reduction in blood glucose levels) [38] and not with metformin (anti-diabetic drugs).

### 3.3. Coriandrum sativum L. (Cilantro)

*Coriandrum sativum* L. is an herbaceous plant widely cultivated in America, Europe, and Eastern countries. In Mexico, it is grown in Puebla, Baja California, Chiapas, Oaxaca, Veracruz, Puebla, Hidalgo, Jalisco, Michoacán, Zacatecas, and Sonora [44]. It belongs to the *Apiaceae* family and is colloquially known as *cilantro* [45] (Figure 4).

The entire plant is edible (leaves and seeds), and it is commonly used as a condiment in different dishes [47].

Various investigations have been carried out to evaluate its hyperglycemic effect and its possible relationship with insulin secretion, as well as body weight control (Table 3). The hypoglycemic effect of the aqueous extract of *Coriandrum sativum* L. seeds, incorporated into the diet (62 ± 5 g/kg) of mice, showed a decrease in blood glucose levels (41%). In addition, extracts using hexane and water, administered at 1 mg/mL in the BRIN–BD11 cell line (rat pancreatic β-cells), presented a higher insulin secretion (50%). On the other hand, the mice in the diabetic group presented a reduction in body weight (21%), while the treated with the extract increased their body weight (4%) compared to the diabetic group (Table 3) [48].

In another investigation, the aqueous extract of *Coriandrum sativum* L. seeds (20 mg/kg) was administered to normal and obese (hyperglycemic) rats with limited physical activity (Table 3). In obese rats, glucose levels were reduced (20%), as well as insulin levels (50%), improving insulin resistance. Moreover, obese rats treated with the extract presented a decrease in body weight (8%) after 30 days. Moreover, the oral administration of the extract caused a reduction in plasma total cholesterol (48%), HDL (28%) and LDL (55%) [49].

In another study, the ethanolic extract of *Coriandrum sativum* L. seeds, using doses of 100, 200, and 250 mg/kg, was investigated in diabetic rats (Table 3). Doses of 200 and 250 mg/kg reported a reduction in blood glucose (33%), as well as an increase in active pancreatic β cells compared to their diabetic control, improving insulin secretion [50].

Recently, the hypoglycemic effect of polyphenols (extracted with methanol) contained in the seeds of *Coriandrum sativum* L. in an in vivo diabetic model was evaluated, with doses of 25 and 50 mg/kg. The results showed hypoglycemic effects (50%) in addition to improving complications associated with T2D, such as dyslipidemia and body weight loss (48% and 40%, respectively) [51].

The effect of the seed powder of *Coriandrum sativum* L. in diabetic and normoglycemic rats, with a dose of 10 g of powder/100 g of food, produced a lower postprandial glucose concentration (44%) and a reduction in fat accumulation, as well as an increase in plasma insulin (40%) in the diabetic group. This may be due to the high content of antioxidants which helps the preservation of pancreatic β-cells [52]. The aqueous extract of *Coriandrum sativum* L. seeds at different concentrations (250 and 500 mg/kg) (Table 3) demonstrated a greater reduction in blood glucose (49%), as well as in cholesterol levels and an increased high-density lipid cholesterol level in a dose of 500 mg/kg [53].

In diabetic rats where an ethanolic extract of *Coriandrum sativum* L. leaves was administered at a dose of 100 mg/kg, a reduction in glucose (49%) and triglyceride levels was obtained. An improvement in the histopathology of pancreatic β cells was observed, as well as a restoration in body weight; however, insulin secretion was not evaluated (Table 3) [54].

The effects of *Coriandrum sativum* L. extract were evaluated compared to glibenclamide, demonstrating a similar hypoglycemic effect (20%) and improvement in insulin resistance [49]. Moreover, in another study, blood glucose levels were evaluated by comparing the effects of the plant (49% reduction with a dose of 40 mg/kg) with metformin (61% reduction), but neither insulin secretion nor body weight was evaluated in this study [55].

After reviewing the articles referring to this plant, it was observed that *Coriandrum sativum* L. could present a decrease in blood glucose and lipids levels, an increase in insulin sensitivity and secretion, an improvement in β cells, and restore body weight. These effects can be due to the chemical compounds it presents (quercetin and chlorogenic acid quercetin (Table 6).

### 3.4. Persea americana Mil. (Aguacate)

*Persea americana* Mil. is a tree native to Central and South America. In Mexico, it is grown in Baja California Sur, Sonora, Michoacán, Jalisco, the State of Mexico, Hidalgo, Veracruz, Puebla, Tabasco, Oaxaca, Morelos, and Guerrero [56]. It belongs to the *Lauraceae* family and is colloquially known as *aguacate* [57] (Figure 5).

The most common edible part of this plant is the pulp; however, the leaves are used as a spice [57].

The aqueous and methanolic extract of *Persea americana* Mil. leaves evaluated in rats with hypercholesterolemia, using a dose of 10 mg/kg (Table 4), indicated a glucose reduction of 16% (aqueous extract) and 11% (methanolic extract), possibly due to an increase in insulin secretion. In these experiments, there were no significant changes in body weight [59]. Ethanolic extract of *Persea americana* Mil. leaves were evaluated in diabetic mice, with doses of 0.490, 0.980, and 1.960 g/kg, showing a greater reduction in glucose levels (64.27%) with the highest dose, suggesting that the hypoglycemic effects of the extract may be due to an increase in insulin secretion, probably by the chemical compounds it contains (flavonoids, saponins, triterpenes); however, in these articles, insulin secretions were not measured (Table 4) [60]. Another group evaluated the effect of the aqueous, ethanolic, and methanolic extract of *Persea americana* Mil. leaves in diabetic rats with a dose of 100 mg/kg (Table 4). The diabetic rats treated with the extract showed an increase in body weight (15.22%). A decrease in glucose levels of 16.3% (aqueous extract), 20.8% (ethanolic extract), and 37.4% (methanolic extract) and recovery of the islets of Langerhans were also observed. Better results were obtained in terms of the hypoglycemic effects of the methanolic extract resembling metformin [61]. Moreover, the hydroalcoholic extract (50% ethanol) of the leaves of *Persea americana* Mil. administered in diabetic and normoglycemic rats, with doses of 0.15 and 0.3 g/kg, produced a reduction in glucose levels of 60% and 71%, respectively (Table 4). In addition, activation of protein kinase B (PKB/Akt) was obtained in the liver and skeletal muscle. In this study, an improvement in pancreatic β cell histology was also obtained; however, it did not show changes in insulin levels. The dose of 0.3 g/kg presented a higher body mass gain compared to the diabetic group, and a lower food intake was also observed [62].

On the other hand, the aqueous extract of the seed of *Persea americana* Mil. was studied in diabetic and normoglycemic rats, using a dose of 300 and 600 mg/kg, showing a reduction in blood glucose of 73.23% and 78.24%, respectively. Moreover, this extract showed a protective effect of pancreatic β cells; however, insulin levels were not measured [63]. In another study, where the aqueous extract of *Persea americana* Mil. seed was also evaluated, they demonstrated a significant reduction in blood glucose levels (51%) and an increase in body weight (12%) in diabetic rats, with doses of 400, 800, 1200 mg/kg (Table 4) [64].

The administration of the hot water extract of *Persea americana* Mil. seeds was investigated in diabetic rats at doses of 20, 30, and 40 g/L (Table 4), showing a hypoglycemic effect similar to gibenclamide, with a decrease of 58.9% of blood glucose levels. During the 21 days of the experiment, the body weight of diabetic rats was reduced; however, the administration of the extract restored the body weight loss to normal. In addition, histopathological studies showed a protective and restorative effect on the liver, pancreas, and kidney tissues [65]. Subsequently, another study evaluated the aqueous and methanolic extract of *Persea americana* Mil. seed in diabetic rats, with doses of 200 and 300 mg/kg (Table 4). The aqueous and methanolic extracts presented an increase in body weight of 7.4% and 21.5%, respectively, compared with the control diabetic group (without the extract). A decrease in glucose levels and improvement in liver function were also obtained, with favorable results with the methanolic extract at a dose of 200 mg/kg (glucose reduction of 70.7%) having similar effects to insulin administration [66]. Recently, the hypoglycemic effect of the ethanolic extract of *Persea americana* Mil. seeds was studied in diabetic rats, with doses of 300, 600, and 1200 mg/kg (Table 4), obtaining a decrease of 50% in blood glucose levels, similar to the effects with glibenclamide [67].

In another study, ethanolic extract from *Persea americana* Mil. pulp was investigated in diabetic rats at a dose of 300 mg/kg (Table 4). It was observed that the levels of glucose, body weight, glycosylated hemoglobin, urea, serum creatinine, and insulin in plasma reverted almost to normal levels [68].

On the other hand, avocatin B, a compound exclusively found in avocados (in seeds, pulp, and peel), can act as an inhibitor of fatty acid oxidation. Therefore, its influence on improving insulin resistance was evaluated in an animal model of obese rats (100 mg/kg), in a β cell line (INS-1) at a dose of 25 μM, and in a skeletal muscle cell line (C2C12) with a dose of 25 μM (Table 4). The results showed an improvement in insulin resistance and an increase in the secretion of insulin (40%) in INS-1 cells, a reduction in glucose levels in obese mice (20%), as well as a decreased in body weight (25%) the supplemented obese mice between baseline and day 30 of treatment. Moreover, an enhancement of glucose uptake by C2C12 cells was observed. On the other hand, a pilot study was carried out with humans (n = 10), administering 50 mg or 200 mg per day for 60 days of avocatin B, observing that its consumption had no effect on blood markers of kidney, liver, and muscle toxicity (total bilirubin, alanine aminotransferase, creatinine, and creatine phosphokinase), concluding that the supplementation with avocatin B does not generate discomfort or harmful effects. Insulin secretion and glucose levels were not measured in humans because the objective was to assert avocatin B toxicity in humans. However, it was reported that 200 mg of avocatin B per day reduced body weight by 6% [69].

In the articles reviewed of this plant, the different extracts were compared primarily with glibenclamide having a similar hypoglycemic effect (58.9%), restoring body weight loss and a protective effect in the pancreas [65]. On the other hand, there have been studies where the hypoglycemic effects of the plant (reduction of 79% with a dose of 53.3 mg/kg) have been compared with metformin (reduction of 80%) [70].

After reviewing the articles referring to this plant, it was observed that *Persea americana* Mil. can present a decrease in blood glucose and lipids levels, an increase in insulin sensitivity and secretion, an improvement in β cells, and restore body weight. These effects can be due to the chemical compounds it presents (avocatin B and saponins) (Table 6).

### 3.5. Bidens pilosa (Amor seco)

*Bidens pilosa* is a medicinal plant belonging to the *Asteraceae* family, colloquially known as *amor seco*, native to America [71]. In Mexico, it is cultivated in Aguascalientes, Durango, Guanajuato, Guerrero, Hidalgo, Jalisco, Michoacán, Morelos, Puebla, Querétaro, Tabasco, Tlaxcala, Veracruz, and Zacatecas [72]. It is widely used as a herbal remedy to treat diseases such as T2D and type 1 diabetes, leukemia, ulcers, gastric neoplasia, and inflammation [73] (Figure 6).

This plant is widely used as fodder for domestic animals. It also has an edible flower that is used in salads; the shoots and leaves are used as potherbs [71].

Its antidiabetic effects, as well as its protective effect on pancreatic β cells (Table 5), have been investigated using extracts from the leaves, flowers, and seeds of *Bidens pilosa* in both in vivo and in vitro models.

The antidiabetic effect of the aqueous extract of the whole plant of *Bidens pilosa* was evaluated in vivo (C57BL/KsJ-db/db mice), with doses of 10, 50, and 250 mg/kg (Table 5). A greater decrease in glucose (50%) and increase in serum insulin (40%) was observed in a dose-dependent manner. Furthermore, the effect of this extract on glucose tolerance was evaluated in db/db mice with a dose of 50 mg/kg, resulting in an improvement in glucose tolerance, such as glibenclamide. A 0.9% reduction in glycated hemoglobin was also obtained, and in histological studies of the pancreas, the extract showed a protective effect on pancreatic β cells. Moreover, the group treated with the extract presented a decrease in body weight (13%) compared with the control group [75]. In another study, the aqueous ethanol extract from the stem of *Bidens pilosa* (1 g/kg) was investigated in C5BL/Ks-db/db mice (T2D model), obtaining blood glucose reduction (33%) (Table 5). In addition, the polyenes mixture presented a dose-dependent reduction in blood glucose levels for 0.5 g/kg (41%) compared to 0.25 mg/kg (25%). However, its effect on insulin secretion was not investigated in this article. On the other hand, no significant differences in body weight were observed; however, the administration of the polyenes mixture reduced food intake [76]. In a more recent study, the hypoglycemic effect of the aqueous extract of *Bidens pilosa* branches was evaluated in diabetic rats (doses of 70, 90, and 110 mg/mL) and normoglycemic rats (doses of 90 mg/mL) (Table 5). A significant reduction in blood glucose levels (18.4%) was obtained with a dose of 70 mg/mL in the diabetic group and a dose of 90 mg/mL in normoglycemic rats (33.58% reduction). Insulin levels were not evaluated in this article; however, it is suggested that it can improve insulin secretion due to the compounds it contains (saponins, tannins, and polyphenols) [77].

A polyacetylene compound from *Bidens pilosa*, called cytopiloyne, was administered in db/db mouse model with doses of 0.1, 0.5, and 2.5 mg/kg (Table 5). This compound exhibited a reduction in postprandial blood glucose levels (61%), an increase in blood insulin (54%), an improvement in glucose tolerance, a reduction in glycosylated hemoglobin levels, and protection of pancreatic β cells. To learn more about the mechanism of action of cytopiloyne, we evaluated its effect on insulin mRNA transcription in the RIN-m5F cell line (pancreatic β cells) at concentrations of 7, 14, and 28 µM. Insulin transcript increased two-fold at a dose of 28 μM cytopiloyne, as well as the intracellular insulin concentration. Moreover, an increase in the mobilization of calcium and diacylglycerols was demonstrated. Cytopiloyne increased PKC-α activation, which is involved in insulin secretion from pancreatic β cells and is activated by diacylglycerols and calcium. This indicated that cytopiloyne enhances β-cell insulin secretion and expression through the regulation of PKC-α by calcium and diacylglycerols [78].

The hypoglycemic effect of *Bidens pilosa* has also been tested in a pilot study in diabetic and normoglycemic humans (n = 10) with a dose of 400 mg (Table 5). A decrease in fasting glucose levels (39%) was observed, as well as glycated hemoglobin levels and increased serum insulin (20%) in normoglycemic subjects. Moreover, it was observed that the mixture of the formulation of *Bidens pilosa* and antidiabetic drugs caused better glycemic control in diabetic patients. These results could be due to an improvement in the function of pancreatic β cells [79].

In the articles reviewed for this plant, the different extracts were compared primarily with glibenclamide, having a similar hypoglycemic effect (50%) [75]. On the other hand, there have not been studies where it has been compared with metformin.

After reviewing the articles referring to this plant, it was observed that *Bidens pilosa* could present a lowering in blood glucose, an increase in insulin secretion, and glucose tolerance. These effects can be due to the chemical compounds it presents (cytopiloyne and tanins) (Table 6).

**Table 6 nutrients-15-02070-t006:** Summary of the effects, compounds, and toxicity of the five reviewed plants. ↓ Decrease. ↑ Increase.

Plant	*Momordica charantia* L. (*Melón amargo*)	*Cucurbita ficifolia bouché*. (*Chilacayote*)	*Coriandrum sativum* L. (*Cilantro*)	*Persea americana* Mil. (*Aguacate*)	*Bidens pilosa*. (*Amor seco*)
**Effects**	↓ Glucose levels ↑ Insulin secretion ↑ Insulin sensitivity ↑ Enhancement of pancreatic β cells ↓ Lipids (serum) ↑ Restore body weight	↓ Glucose levels ↑ Insulin secretion ↑ Enhancement of pancreatic β cells ↓ Lipids (serum) ↑ Restore body weight	↓ Glucose levels ↑ Insulin secretion ↑ Insulin sensitivity ↑ Enhancement of pancreatic β cells ↓ Lipids (serum) ↑ Restore body weight	↓ Glucose levels ↑ Insulin secretion ↑ Insulin sensitivity ↑ Enhancement of pancreatic β cells ↓ Lipids (serum) ↑ Restore body weight	↓ Glucose levels ↑ Insulin secretion ↑ Glucose tolerance ↑ Enhancement of pancreatic β cells ↑ Restore body weight
**Compounds involved.**	- Saponins - Proteins - Triterpenes - Polysaccharides	- D-quiro-inositol - Phenolic compounds (gallic acid, chlorogenic acid)	- Antioxidants (quercetin) - Polyphenols (chlorogenic acid quercetin)	- Avocatin B - Saponins	- Cytopiloyne - Polyphenols (tannins)
**Toxicity**	- The proteins, triterpenes, and polysaccharides of *Momordica charantia* L. have not shown toxicity - The toxicity of saponins from *Momordica charantia* L. has not been evaluated. However, an LD 50 (oral, rat) of 960 mg/kg is recommended	- Gallic acid: LD50 (oral, rat) 5000 mg/kg - Chlorogenic acid: it does not present toxicity -Inositol: LD50 (oral, mouse) 10,000 mg/kg	- Quercetin: LD50 (oral, rat) 161 mg/kg LD50 (oral, mouse) 160 mg/kg - Chlorogenic acid: it does not present toxicity	- Avocatin B: the oral dose of 100 mg/kg in rats does not present toxicity In humans, the 200 mg dose is not toxic either - Persin: LD50 of 751.6 mg/kg administered intraperitoneally in rats - The toxicity of saponins from *Persea americana* Mil. has not been evaluated. However, it is recommended an LD 50 (oral, rat) of 960 mg/kg	- Tannins: LD50 (oral, rat) 2260 mg/kg - Cytopiloyne: at doses of 2.5 mg/kg orally in mice, they do not present toxicity

## 4. Discussion

Mexican plants have been used for many centuries for the treatment of T2D and, recently, for body weight control. In México, the Federal Commission for the Protection against Sanitary Risk (COFEPRIS) allowed the use of 18 medicinal plants as herbal medicine [80]. For example, *Echinacea purpurea* is approved in Mexico, Belgium, Bulgaria, Croatia, and Denmark with a dose of 100 mg [81]. Another example is *Panax ginseg*, which is approved in Mexico, Austria, Belgium, Denmark, France, Germany, Ireland, Portugal, and Spain with a dose of 100 mg or 300 mg [82]. Moreover, *Passiflora incarnata* is also approved in Mexico, Germany, Austria, and Belgium with a dose of 200 mg [83]. For herbal compounds commercialization, the advertising and marketing must include therapeutic indication and symptomatic effect; in no case could these products be advertised as curatives [84].

However, there are many other medicinal plants that have scarce scientific studies demonstrating the physiological and molecular mechanism of action of these plants over glucose homeostasis and body weight. Therefore, the need to review the literature and describe the state of the art of these plants is necessary to ultimately propose future routes to study their mechanism of action and their plausible use to treat these diseases.

### 4.1. Momordica charantia L. (Melón amargo)

When reviewing the articles referring to *Momordica charantia* L., its antidiabetic effects may be due to the chemical compounds present in its aqueous or organic extracts such as proteins [27] saponins [28,29], triterpenes [30], and polysaccharides [29] (Table 2).

The proteins contained in *Momordica charantia* L. were found to increase plasma insulin levels and glucose uptake, suggesting that it may also have an effect on improving insulin sensitivity. However, the type of proteins that were extracted from *Momordica charantia* L. was not mentioned; therefore, more studies are needed to identify the active compounds that promote the beneficial effects mentioned above [27]. Regarding saponins, their hypoglycemic action is caused by the improvement of insulin response [29], increasing insulin levels, and improving pancreatic β-cell function [28]. However, it is important to consider that saponins can present toxicity and irritability in the intestine when ingested in high amounts (lethal dose (LD) 960 mg/kg) [85] (Table 2).

T2D has been shown to be associated with increased free radical formation and decreased antioxidant potential, leading to oxidative stress induced by hyperglycemia and free fatty acids, causing insulin resistance and pancreatic β-cell dysfunction [86]. In the articles reviewed, both the saponins [28] and the triterpenes [30] of *Momordica charantia* L. presented an antioxidant effect improving and preserving pancreatic β cells; this is in conjunction with its hypoglycemic action and increased insulin production.

It is important to highlight that in the reviewed articles, it was also observed that the extracts used from this plant had similar effects to the pharmacological controls [24] with glibenclamide, a sulfonylurea that increases the secretion of insulin by stimulating pancreatic β cells [87]. Therefore, it is suggested that this plant could present a mechanism of action similar to sulfonylureas. On the other hand, it has been investigated that the inhibition of PTP-1B plays an important role in the signaling of metabolic pathways and that it may be a possible therapeutic target for T2D. This protein participates in the negative regulation of insulin receptor signaling [88]. In one of the reviewed articles, it is reported that *Momordica charantia* L. triterpenes exerted an inhibiting effect on PTP-1B, improving the effects caused by insulin resistance [30], which may represent a possible treatment for T2D.

Moreover, *Momordica charantia* L. has the effect of restoring body weight, which could be a consequence of its positive influence on lipid metabolism (decrease in serum triglycerides, cholesterol, and LDL levels) [22,23,25,28,29]. On the other hand, one of the articles reviewed mentioned that the extract of this plant could reduce fat mass and increases SIRT1 levels, which can increase hepatic gluconeogenesis and fatty acid use, contributing to the loss of fat mass [26]. In addition, these effects have been associated with the inhibition of pancreatic lipase (which breaks down fats so they can be absorbed in the intestines) or modulating appetite, influencing weight gain or loss [89].

In addition, *Momordica charantia* L. has the effect of restoring body weight, decreasing blood glucose and lipids levels, increasing insulin secretion, improving insulin sensitivity, and the histology of pancreatic β cells (Table 2). Therefore, this plant is a good candidate to further study its beneficial effects on glucose homeostasis and body weight control.

### 4.2. Cucurbita ficifolia bouché (Chilacayote)

*Cucurbita ficifolia bouché* presented similar effects to *Momordica charantia* L. because they belong to the same family (*Cucurbitaceae*). Both plants can reduce blood glucose levels [37,38,39,40,41], increase the concentration of insulin [35,40,41,42,43], and improve the functioning of pancreatic β cells [41]. Moreover, *Cucurbita ficifolia bouché* increases intracellular calcium concentration in in vitro models [43] (Table 2).

*Cucurbita ficifolia bouché* contains phenolic compounds, mainly gallic acid, chlorogenic acid, and quercetin (Table 6). These compounds have an antioxidant effect, preventing the death of pancreatic β cells and allowing the recovery of those partially destroyed [41]. On the other hand, it has been reported that phenolic compounds can increase insulin secretion by stimulating GLP-1 (incretin) secretion [90]. Incretin hormones are peptides released from the gastrointestinal tract in response to food ingestion. These hormones enhance insulin secretion and regulate glucose homeostasis [91]. Moreover, gallic acid has been reported to reduce glucose levels in diabetic rats due to its insulin-releasing effect [92]. On the other hand, chlorogenic acid has been associated with an increase in insulin sensitivity, improvement in glucose tolerance, and an increase in insulin secretion [93]. Therefore, these compounds can potentiate the hypoglycemic effect of *Cucurbita ficifolia bouché* by different mechanisms.

On the other hand, it has been suggested that the active principle of this plant may be the compound called D-quiro-inositol. Inositol molecules, particularly D-quiro-inositol, have been suggested to be important mediators of insulin action [94]. D-quiro-inositol is normally present in the urine and blood, but it is at greatly reduced levels or absent in T2D patients, which has been associated with a decrease in insulin sensitivity [95]. The possible mechanism of action of this compound has been investigated using aqueous extracts from the fruit of *Cucurbita ficifolia bouché* [42,43]. These studies proposed that its hypoglycemic effects may be mediated by an increase in the expression of mRNA for insulin and for the major subunit of the ATP-sensitive K⁺ channel (Kir6.2), a protein present in pancreatic β-cells, which plays an important role in insulin secretion [43] and mediate the increase in intracellular calcium and insulin secretion [43] It is important to consider that D-quiro-inositol is found in a higher concentration, mainly in the ripe fruit; however, hypoglycemic capacity has also been shown in diabetic patients with the immature fruit used in doses of 4 mL/kg [37], as well as when using immature fruit seeds in diabetic mouse models (dose of 500 mg/kg) [39]. D-quiro-inositol has also been associated with body weight control by reducing BMI in obese patients [96] and restoring body weight during T2D, which can be due to its positive effects in insulin secretion and hypoglycemic effects [41]. Therefore, this suggests that the hypoglycemic effect of *Cucurbita ficifolia bouché* not only depends on D-quiro-inositol but also on other compounds, such as phenolics, which are found in both ripe and unripe fruit [39]. On the other hand, D-quiro-inositol has an oral LD50 toxicity (mouse) of 10,000 mg/kg [97] (Table 6).

For many of the articles reported on this plant, the antidiabetic effect of *Cucurbita ficifolia bouché* was compared with pharmacological controls such as tolbutamide (sulfonylurea). This drug acts on the pancreas, stimulating functional pancreatic β cells and causing an increase in insulin secretion [98]. Considering the above, the extracts of *Cucurbita ficifolia bouché* presented similar results to those observed with tolbutamide (stimulates the secretion of insulin by the pancreas) [40]. This suggests that they might have a similar mechanism of action.

On the other hand, it is important to consider the toxicity of the compounds of this plant, which are mainly: gallic acid (LD50 (oral, rat) 5000 mg/kg), chlorogenic acid (LD50 (intraperitoneal, rat) 4000 mg/kg), and inositol (LD50 (oral, mouse) 10,000 mg/kg) (Table 6).

### 4.3. Coriandrum sativum L. (Cilantro)

*Coriandrum sativum* L., such as *Momordica charantia* L. and *Cucurbita ficifolia bouché*, presented an improvement in pancreatic β cells [50], which can be attributed to its content of antioxidants, especially the flavonoid quercetin (a phenolic compound) present in plants and foods [52] (Table 6).

Quercetin supplementation has been shown to help normalize blood glucose concentration [99]. In addition, it has been shown to improve antioxidant status [50], prevent oxidative damage, help pancreatic islet regeneration, and increase insulin secretion [50]. Quercetin can be obtained from the powder of its seeds [52]. Coriander does not present high toxicity; however, quercetin has an LD50 (oral, rat) of 161 mg/kg (Table 6) [52]. Therefore, it would be important to know the concentration of this compound in coriander to determine a dose of extract that does not represent toxicity for animal models or humans. *Coriandrum sativum* L. also has polyphenols, especially chlorogenic acid, which may be responsible for its hypoglycemic effect [51] since it has been associated with an increase in insulin sensitivity, improvement in glucose tolerance, and increase in insulin secretion [93]. On the other hand, one of the reviewed articles suggests that *Coriandrum sativum* L. improves insulin sensitivity [49], similar to *Momordica charantia* L. [30]. However, it is important to consider that although *Coriandrum sativum* L. extract was able to lower blood glucose and insulin resistance indicators in obese rats, they were subjected to physical activity. This could be a variable that could have influenced the hypoglycemic results of the extract [49]. It is also important to consider that the effects of *Coriandrum sativum* L. in increasing insulin concentration [50] are dose-dependent. At a dose of 20 mg/kg, it does not present a significant increase in insulin secretion [49], while at higher doses (200 mg/kg), the concentration of this hormone significantly increases [50].

On the other hand, *Coriandrum sativum* L. restored body weight loss during diabetes, which can be because of its ability to improve insulin sensitivity and decrease lipids and glucose levels [48,51,52]. In addition, this plant has effects on weight loss in an obese animal model; this can be due to its chemical compounds, such as quercetin [49]. This compound has been associated with a decrease in body weight, food intake, plasma cholesterol, and fat accumulation [100].

Considering the previous studies, we can suggest that the antidiabetic effects of *Coriandrum sativum* L. can mainly be explained by the protective effect on pancreatic β cells of the flavonoids and polyphenols present in the seeds, causing an improvement in insulin secretion. However, it is important to consider that toxicity, in particular of quercetin, presents an LD50 (oral, rat) 161 mg/kg and an LD50 (oral, mouse) 160 mg/kg, while chlorogenic acid has an LD50 (intraperitoneal, rat) 4000 mg/kg (Table 6).

### 4.4. Persea americana Mil. (Aguacate)

*Persea americana* Mil. can cause a hypoglycemic effect [60,62,63,64,65,67,68], as well as an increase in insulin sensitivity [69], an improvement in pancreatic β-cell function and histology [63,65,66], and an increase in insulin levels [68,69] (Table 4).

Studies have suggested that flavonoids, present in *Persea americana* Mil. could produce a decrease in serum glucose levels and an increase in glucose uptake by peripheral tissues via protein kinase B (PKB/Akt) [101,102]. Activation of this enzyme can increase glucose transport by translocating GLUT-4 from the cytosol to the plasma membrane, thereby increasing glucose uptake in skeletal muscle, adipocytes, liver, and other tissues [103].

On the other hand, a compound present in *Persea americana* Mil., called avocatin B, has recently been investigated. Avocatin B is a mixture of avocadene and avocadin, two 17-carbon polyhydric fatty alcohols originally discovered in avocado seeds. However, it has been reported that up to 200 mg of avocatin B may be present in avocado pulp [69]. This compound is capable of increasing insulin production both in vivo (obese rats) and in vitro (INS-1 cells). Although little is known about its mechanism of action, it has been proposed that it is capable of inhibiting the oxidation of fatty acids, which in turn causes an increase in the use of glucose and a decrease in insulin resistance [69].

In the reviewed articles, *Persea americana* Mil. restored the body weight loss caused by diabetes [61,62,64,65,66,68]. In one of the articles, it was mentioned that this plant could also cause body weight loss in a model of obese mice; this may be due to the benefits that avocatin B could present in glucose and fat metabolism [69].

On the other hand, it is important to consider that toxicological studies in which *Persea americana* Mil. pulp has been administered orally have established an LD50 of 15,000 mg/kg in rats. This effect may be related to the presence in *Persea americana* Mil. seeds and leaves of a compound called persin, which has a toxic effect on rats and mice, in whom an LD50 of 751.6 mg/kg administered intraperitoneally is reported [66]. Persin is also found in *Persea americana* Mil. pulp but in a lower concentration, and although a toxic effect on humans has not been demonstrated [69], it is important to consider its toxicity to avoid an adverse effect in the future experiments.

Therefore, the hypoglycemic effect of avocado may be due to the stimulating effect on pancreatic β-cells, making them capable of secreting more insulin, or to a protective effect on pancreatic β-cells. This may be due to the chemical compounds present mainly in its leaves (flavonoids) and pulp (avocatin B) (Table 6). In addition, it is important to consider that avocatin B, both at high doses (1200 mg/kg) and at lower doses (10 mg/kg), produces a decrease in blood glucose levels and has not shown toxic effects. Additionally, it is important to consider that *Persea americana* Mil. has effects on insulin secretion in both in vivo and in vitro models.

### 4.5. Bidens pilosa (Amor seco)

*Bidens pilosa* can lower blood glucose levels [76,77,78,79] and increase insulin concentration [75,79]. In addition, it improves glucose tolerance and pancreatic β-cell function [75,78] (Table 5).

This plant has polyphenolic compounds, in particular flavonoids and tannins. These compounds also exert antioxidant activity, preserving and improving the function of pancreatic β cells [77]. It has also been reported that tannins and flavonoids have the property of slowing down the absorption of sugars in the intestine, which limits and regulates their passage into the blood [90].

*Bidens pilosa* also showed a reduction in glucose levels similar to that shown by glibenclamide (sulfonylurea). Sulfonylureas have been reported to have hypoglycemic effects by stimulating insulin production through pancreatic β cells. *Bidens pilosa* presented a reduction in glucose levels [77] and an increase in serum insulin levels similar to those presented by glibenclamide [75].

Recently, a polyacetylene, called cytopiloyne, present in *Bidens pilosa* has been identified [78]. It has been observed that this compound is capable of stimulating insulin secretion. Its mechanism of action is not yet fully understood. However, cytopiloyne has been proposed to improve pancreatic β-cell function and intracellular calcium flux, thereby enhancing insulin synthesis and secretion [104]. Cytopiloyne has also been proposed to increase PKC-α activation, which enhances insulin secretion and expression in pancreatic β cells [78].

*Bidens pilosa* also presented the effect on body weight loss in a model of db/db mice [75], probably due to its chemical compounds, specifically its polyenes [76]. Polyenes have been associated with a reduction in fat accumulation, adipocyte size and/or body weight [105].

It is worth mentioning that, as in the case of *Cucurbita ficifolia bouché*, experiments have been carried out in diabetic patients using a formulation of *Bidens pilosa*, presenting hypoglycemic effects and an increase in insulin secretion. However, this study was performed on a small population of 10 patients. In addition, the composition and elaboration of the plant formulation are not mentioned [79].

On the other hand, it must be considered that tannins present an LD50 (oral, rat) of 2260 mg/kg [106]. As for cytopiloine, it has been observed that in doses of 2.5 mg/kg orally in mice, it does not present toxicity [78] (Table 6).

## 5. Conclusions

In this review, five Mexican medicinal plants used to treat T2D were selected. It was observed that both aqueous and organic extracts of different plants could have hypoglycemic effects, increasing insulin secretion and sensitivity, improving the function of pancreatic β cells and glucose tolerance, as well as regulating body weight. These effects may be due to the different chemical compounds present in these plants, such as saponins, phenolic compounds, antioxidants, tannins, triterpenes, avocatin B, cytopiloyne, etc. However, it is important to further study the toxicology of the different plants and their chemical compounds to determine the recommended doses for their hypoglycemic and body weight control effect.

However, for most of these plants, the mechanism of action by which they exert their hypoglycemic effect is not fully understood. Therefore, it is necessary to further study the different extracts in both in vivo and in vitro models to continue validating the evidence for its implication in glucose homeostasis and body weight control.

In addition, it is important to investigate the active principle of these plants that helps modulate glucose homeostasis, insulin secretion, and body weight control, as well as its mechanism of action. In the future, these plants could be proposed as a possible treatment for T2D and obesity, impacting a more economical and accessible option for patients with these diseases.

## Figures and Tables

**Figure 1 nutrients-15-02070-f001:**
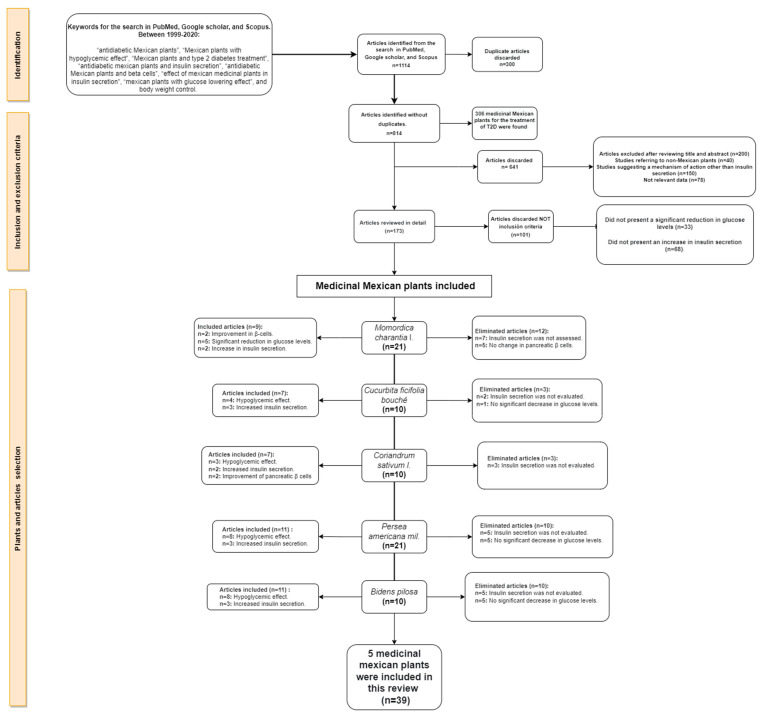
Systematic review flowchart. Identification, eligibility, plants selection, and analysis, using PRISMA (Preferred Reporting Items for Systematic Reviews and Meta-Analysis).

**Figure 2 nutrients-15-02070-f002:**
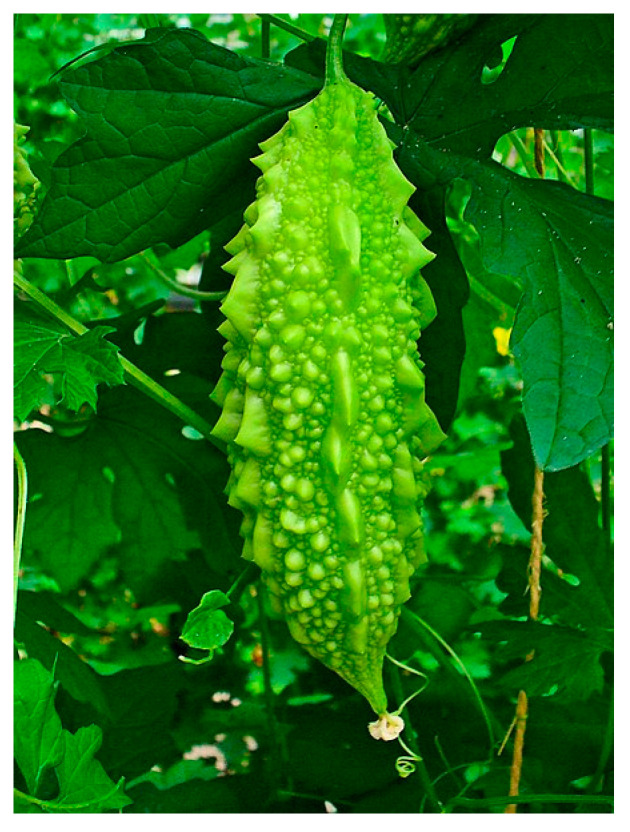
*Momordica charantia* L. (*melón amargo*) plant. Published by Wikimedia Commons licensed by Creative Commons, reprinted from [21]. https://creativecommons.org/licenses/by-sa/3.0/legalcode, accessed on 10 March 2023.

**Figure 3 nutrients-15-02070-f003:**
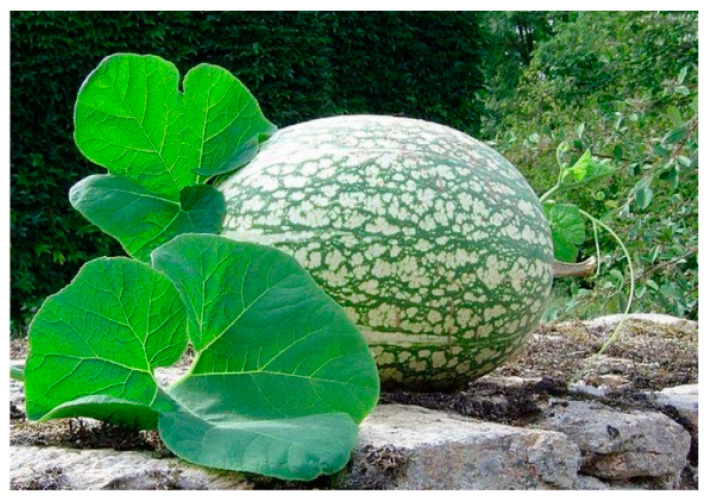
*Cucurbita ficifolia bouché* (*chilacayote*) plant. Published by Wikimedia Commons licensed by Creative Commons, reprinted from [36]. https://creativecommons.org/licenses/by-sa/3.0/legalcode, accessed on 10 March 2023.

**Figure 4 nutrients-15-02070-f004:**
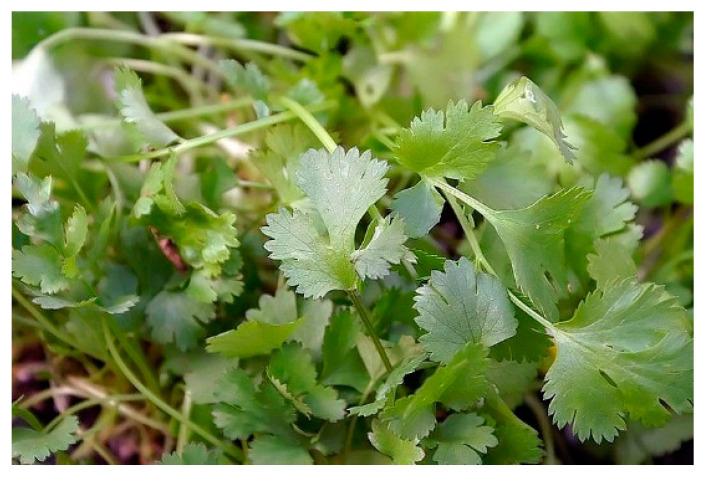
*Coriandrum sativum* L. (*cilantro*). Published by Wikimedia Commons, licensed by Creative Commons, reprinted from [46]. https://creativecommons.org/licenses/by-sa/4.0/legalcode, accessed on 10 March 2023.

**Figure 5 nutrients-15-02070-f005:**
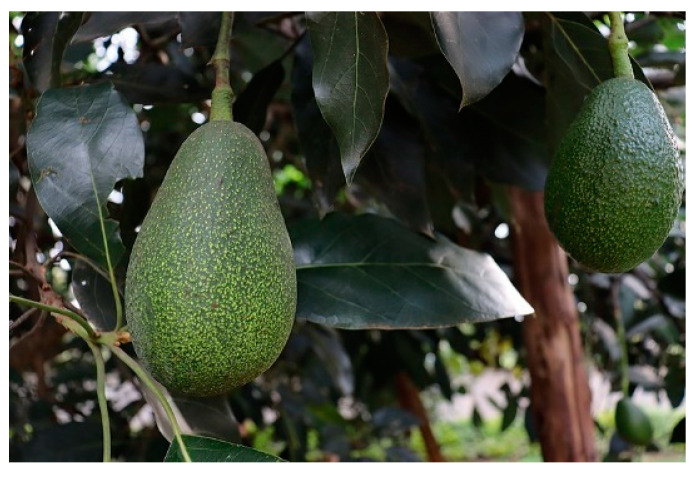
*Persea americana* Mil. *(aguacate*) plant. Reprinted from [58].

**Figure 6 nutrients-15-02070-f006:**
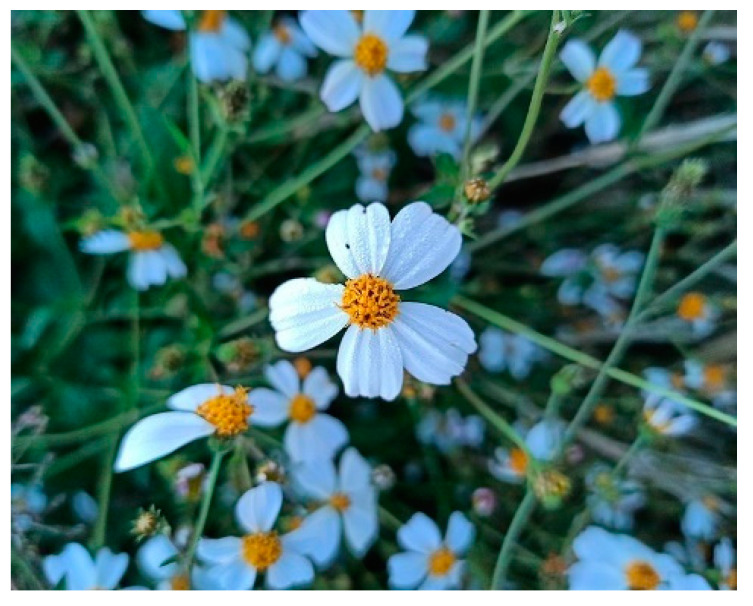
*Bidens pilosa* (*amor seco*) plant. Published by Wikimedia Commons, licensed by Creative Commons, reprinted from [74]. https://creativecommons.org/licenses/by-sa/4.0/legalcode, accessed on 10 March 2023.

**Table 1 nutrients-15-02070-t001:** Summary of reviewed articles of *Momordica charantia* L., considering the type of extract, the part of the plant, dose, experimental model, body weight, and physiologic or cellular effects. ↓ Decrease. ↑ Increase.

Plant	Extract	Part of the Plant	Dose	Experimental Model	Physiologic or Cellular Effects	Body Weight	Reference
***Momordica charantia* L.** (***melón amargo***)	Methanol	Leaves	200 mg/kg 400 mg/kg	Wistar diabetic rats (alloxan)	Both doses showed: ↓ Glucose levels (50%) ↓ Lipids ↑ Antioxidant effect ↑ Structure of pancreatic β cells	- Diabetic rats: ↓ Body weight. - The group treated with the extract: ↑ Body weight, reversing the weight loss caused by diabetes	[22]
	Water	Leaves	100 mg/kg 400 mg/kg	Wistar diabetic rats (alloxan)	Both doses showed: ↓ Glucose (21%) ↓ Lipid levels Insulin secretion was not evaluated	- The diabetic group presented: ↓ Body weight loss of 25.11% - The group threatened with the extract presented: ↑ Body weight of 14.33%	[23]
	Water	Fruit	20 mg/kg	Streptozotocin-induced type 2 diabetic neonatal rats	↓ Glucose levels (33.3%) ↑ Serum insulin levels (33%) ↓ Damage to pancreatic islets		[24]
	Ethanol	Fruit	1 mg/kg 3 mg/kg	Normoglycemic and diabetic rabbits	- 1 mg/kg: ↓ Blood glucose levels (54.8%), ↓ Serum insulin levels (13%) - 3 mg/kg: ↓ Blood glucose levels (61.35%) ↑ Serum insulin levels (17%)	- Normoglycemic rabbits: ↓ Body weight of 4.4% - Diabetic rabbits ↓ Body weight of 1.35% - 1 mg/kg: ↓ Body weight 1.19% - 3 mg/kg: ↓ Body weight of 37%	[25]
	Ethanol	Fruit	250 mg/kg 500 mg/kg	C57BL/6 mice fed a high-fat diet.	Both doses showed: ↑ Insulin sensitivity ↓ Insulin levels (30%) ↓ Glucose blood levels (75%)	- 250 mg/kg: ↓ Body weight (14.8%) compared with the mice fed a high-fat diet - 500 mg/kg: ↓ Body weight (25%) compared to the mice fed with a high-fat diet ↑ SIRT1 levels	[26]
	Acid-ethanol	Pulp (proteins)	in vivo: 1 mg/kg 5 mg/kg 10 mg/kg in vitro: 10 µg/mL	Normoglycemic and diabetic Wistar rats. C2C12 myocytes 3T3-L1 adipocytes	- 10 mg/kg: ↓ Glucose levels in diabetic rats (43%) - The extract: ↑ Plasma insulin concentration. - C2C12 cell line: ↑ Glucose uptake (28%) - 3T3L-1 cell line: ↑ Glucose uptake (35%)		[27]
	Water	Saponins (pulp)	400 mg/kg, 200 mg/kg 100 mg/kg	Diabetic Wistar rats	- The three doses presented: ↓ Glucose levels (50%) ↑ Lipid metabolism Regulation of the insulin signaling pathway ↓ Insulin levels (16.6%) Protective effect of pancreatic β cells	- Diabetic group: ↓ body weight compared with normoglycemic rats - Administration of the extract: ↑ Body weight	[28]
	Isopropyl alcohol	Pulp	Saponins: 20 mg/kg, 40 mg/kg, 80 mg/kg Polysaccharides: 500 mg/kg	Diabetic Kunming mice	- The doses of saponins presented: ↓ Glucose levels (10%) ↓ Insulin resistance returning insulin levels to normal ↑ Proportion of p-AMPK ↑ Antioxidant capacity The dose of 80 mg/kg of saponins presented a better effect	The polysaccharides and saponins extract: ↓ Body weight loss that presents during diabetes	[29]
	Water	Triterpenes (pulp)	20 μM 30 μM	FL83B (Mouse cell line of hepatocytes)	The doses used presented: ↑ Glucose uptake ↑ Phosphorylation of IRS-1 ↑ Insulin sensitivity ↑ AMPK activation ↑ inhibition of PTP-1B		[30]

**Table 2 nutrients-15-02070-t002:** Summary of reviewed articles on *Cucurbita ficifolia bouché*, considering the type of extract, the part of the plant, dose, experimental model, body weight, and physiologic or cellular effects. ↓ Decrease. ↑ Increase.

Plant	Extract	Part of the Plant	Dose	Experimental Model	Physiologic or Cellular Effects	Body Weight	Reference
** *Cucurbita ficifolia bouché (chilacayote)* **	Processor	Fruit juice	4 mL/kg	10 patients with T2D	↓ Glucose levels (31%)		[37]
	Chilacayote freeze-dried juice.	Fruit	250 mg/kg 500 mg/kg 750 mg/kg 1000 mg/kg 1250 mg/kg	CD-1 normoglycemic and diabetic mice	All doses showed: ↓ Glucose levels in normoglycemic and diabetic mice (83%) The best dose was 500 mg/kg, without toxic effects		[38]
	Water	Seeds	500 mg/kg	CD-1 normoglycemic and diabetic mice	↓ Glucose levels in all stages of maturity of the fruit It presented a better effect in lowering glucose levels (50%) at 15 days of development of the fruit compared to glibenclamide		[39]
	Methanol	Fruit without seeds	Diabetic: 300 mg/kg 600 mg/kg Normoglycemic 600 mg/kg	Diabetic and normoglycemic Wistar rats	The two doses presented: ↓ Glucose levels in the diabetic group (60%) ↑ Increase in insulin levels (53%)	- The diabetic group presented: ↓ body weight (25%). - 300 mg/kg: It presented no significant difference in body weight. - 600 mg/kg: ↑ body weight (6%).	[40]
	Methanol	Fruit without seeds	300 mg/kg	Diabetic and normoglycemic Wistar rats	↓ Hyperglycemia (12.5%) ↓ Lipid peroxidation in the pancreas ↑ Insulin levels (36%) in the diabetic group ↑ Active pancreatic β cells		[41]
	Water	Fruit without seeds	0.25 µM of the aqueous extract and D-quiro- inositol	RINmF5 cells (mouse pancreatic β cells)	↑ mRNA of *insulin* (80%) and *Kir6.2* genes ↑ Insulin secretion. Glucose levels were not evaluated		[42]
	Water	Fruit without skin and seeds	*C. ficifolia* (72 μg/mL), D-quiro-inositol (400 μM)	RINmF5 cells (mouse pancreatic β cells)	- D-quiro-inositol: neither the concentration of insulin nor calcium increased - Aqueous extract: ↑ Concentration of calcium and insulin (28.5%)		[43]

**Table 3 nutrients-15-02070-t003:** Summary of reviewed articles of *Coriandrum sativum* L., considering the type of extract, the part of the plant, dose, experimental model, body weight, and physiologic or cellular effects. ↓ Decrease. ↑ Increase.

Plant	Extract	Part of the Plant	Dose	Experimental Model	Physiologic or Cellular Effects	Body Weight	Reference
***Coriandrum sativum* L. (*cilantro*)**	Water Water, hexane, ethyl acetate, and methanol for the BRIN–BD11 cell line	Seeds	Mice: the extract was incorporated in the diet (62 ± 5 g/kg) Cell line: 1 mg/mL	Diabetic CD-1 mice and BRIN-BD11 cell line (rat pancreatic β-cells)	The doses used showed: ↓ Blood glucose levels (41%). The aqueous extract, as well as the one made with hexane: ↑ Secretion of insulin (50%) in the BRIN-BD11 cell line	- The diabetic group presented: ↓ Body weight (21%) - The mice treated with the extract: ↓ Body weight (4%)	[48]
	Water	Seeds	20 mg/kg	Normoglycemic and obese (hyperglycemic) rats with limited physical activity	↓ Glucose (20%) ↓ Insulin levels (50%) ↓ Insulin resistance in obese rats The oral administration of the extract caused: ↓ Plasma total cholesterol (48%), HDL (28%), LDL (55%)	The obese rats treated with the extract presented: ↓ Body weight (8%) after 30 days	[49]
	Ethanol	Seeds	100 mg/kg 200 mg/kg 250 mg/kg	Diabetic Wistar rats	-200 and 250 mg/kg: ↓ Blood glucose (33%) ↑ Activity of the pancreatic β cells ↑ Insulin secretion		[50]
	Methanol	Seeds	25 mg/kg 50 mg/kg	Diabetic Wistar rats	Both doses: ↓ Glucose levels (50%) ↓ Dyslipidemia In this article insulin secretion was not evaluated	- 25 mg/kg: ↑ Body weight (48%) - 50 mg/kg: ↑ Body weight (40%)	[51]
	Powder	Seeds	10 g of powder/100 g of food	Diabetic and normoglycemic Wistar rats	↓ Postprandial glucose concentration (44%) ↑ Plasma insulin (40%) in the diabetic group	↓ Fat accumulation	[52]
	Water	Seeds	250 mg/kg 500 mg/kg	Diabetic Wistar rats	Both doses: ↓ Glucose levels (49%), presenting better hypoglycemic effects at the dose of 500 mg/kg The dose of 50 mg/kg: ↓ The total cholesterol level ↑ The high-density lipid cholesterol level		[53]
	Ethanol	Leaves	100 mg/kg	Wistar diabetic rats (alloxan)	↑ Hypoglycemic effect (49%) ↓ triglycerides improvement in the histopathology of pancreatic β cells Insulin secretion was not evaluated	- The diabetic group presented: ↓ Body weight - Administration of the extract: ↑ Body weight	[54]

**Table 4 nutrients-15-02070-t004:** Summary of reviewed articles of *Persea americana* Mill., considering the type of extract, the part of the plant, dose, experimental model, body weight, and physiologic or cellular effects. ↓ Decrease. ↑ Increase.

Plant	Extract	Part of the Plant	Dose	Experimental Model	Physiologic or Cellular Effects	Body Weight	Reference
***Persea americana* Mil. (*aguacate*)**	Water Methanol	Leaves	10 mg/kg	Wistar rats with hypercholesterolemia	↓ Glucose levels with aqueous extract (16%) ↓ Glucose levels with the methanolic extract (11%)	There were no significant changes in body weight	[59]
	Ethanol	Leaves	0.490 g/kg 0.980 g/kg 1.960 g/kg	Diabetic mice	- 1.960 g/kg: ↓ Glucose levels of 64.27% Insulin secretion was not measured		[60]
	Water Ethanol Methanol	Leaves	100 mg/kg	Diabetic Wistar rats	- Aqueous extract: ↓ Glucose levels (16.3%) - Ethanol extract: ↓ Blood glucose levels (20.8%) - Methanol extract: ↓ Glucose levels (37.4%) Recovery of the islets of Langerhans was observed with the three extracts	↑ Body weight (15.22%)	[61]
	Hydroalcoholic extract	Leaves	0.15 g/kg 0.3 g/kg	Diabetic and normoglycemic Wistar rats	The two doses used are presented: ↓ Glucose levels: 0.15 g/kg (60%) 0.3 g/kg (71%) ↑ Activation of protein kinase B (PKB) in liver and skeletal muscle ↑ Pancreatic β-cell was observed. No change in insulin levels	The dose of 0.3 g/kg: ↑ Body mass gain compared to the diabetic group. ↑ Food intake	[62]
	Water	Seed	300 mg/kg 600 mg/kg	Diabetic and normoglycemic Wistar rats	- 300 mg/kg: ↓ Glucose levels (73.23%) - 600 mg/kg: ↓ Glucose levels (78.24%) Protective effect of pancreatic β cells. Insulin secretion was not measured		[63]
	Water	Seed	400 mg/kg 800 mg/kg 1200 mg/kg	Diabetic Wistar rats	The three doses used showed: ↓ Blood glucose levels (51%)	The group threatened with the extract presented: ↑ Body weight (12%) compared with the diabetic untreated group	[64]
	Hot water	Seed	20 g/L 30 g/L 40 g/L	Diabetic Wistar rats	The three doses used are presented: ↑ Hypoglycemic effect similar to glibenclamide (58.9%) ↑ Protective and restorative effect on the pancreas, liver, and kidney tissues	↓ Body weight of diabetic rats The administration of the extract ↑ Body weight loss toward normal	[65]
	Water Methanol	Seed	Aqueous and methanolic extract: 200 mg/kg 300 mg/kg	Diabetic Wistar rats	Both extracts showed: ↓ Glucose levels ↑ Function of the pancreas. Methanolic extract 200 mg/kg: ↓ Glucose levels (70.7%)	↑ Body weight (7.4% aqueous extract or 21.5% methanolic extract) compared to diabetic control group	[66]
	Ethanol	Seeds	300 mg/kg 600 mg/kg 1200 mg/kg	Diabetic Wistar rats	The doses used showed: ↓ Glucose levels (50%) similar to glibenclamide, having a better result with a concentration of 300 mg/kg		[67]
	Ethanol	Pulp	300 mg/kg	Diabetic Wistar rats	Levels of glucose and insulin returned almost to normal levels	The administration of the extract ↑ Body weight	[68]
	Water	Pulp (Avocatin B)	Obese C57BL/6J mice: 100 mg/kg INS-1 cells: 25 µM C2C12 cells: 25 µM Humans (n = 10): 50 mg o 200 mg per day for 60 days	Obese C57BL/6J mice INS-1 cells (pancreatic β cells) C2C12 cells (myocytes) Humans (n = 10)	- Obese mice: ↓ Glucose levels (20%) - INS-1 cells: ↑ Insulin secretion (40%) ↓ insulin resistance - C2C12 cells: ↑ Glucose uptake - Humans: it did not generate discomfort or toxic effects	↓ Body weight (25%) in obese mice ↓ Body weight (6%) in humans	[69]

**Table 5 nutrients-15-02070-t005:** Summary of reviewed articles on *Bidens pilosa*, considering the type of extract, the part of the plant, dose, experimental model, body weight, and physiologic or cellular effects. ↓ Decrease. ↑ Increase.

Plant	Extract	Part of the Plant	Dose	Experimental Model	Physiologic or Cellular Effects	Body Weight	Reference
** *Bidens* ** ***pilosa* (*amor seco*)**	Water	Whole plant	10 mg/kg 50 mg/kg 250 mg/kg	C57BL/KsJ-db/db mice	The doses showed: ↑ Glucose tolerance (50%) ↑ Increased insulin levels (20%) The dose of 50 mg/kg presented the most similar effect to glibenclamide	The group treated with the extract presented: ↓ Body weight (13%)	[75]
	Ethanol– water	Stem	Extract: 1 g/kg Polyenes mixture: 0.5 g/kg and 0.25 g/kg	C5BL/Ks-db/db mice	A dose of 1 g/kg: ↓ Blood glucose levels (33%) ↓ Blood glucose levels with a concentration of 0.5 g/kg (41%) ↓ Blood glucose levels with a concentration of 0.25 g/kg (25%) Insulin secretion was not measured	No significant differences in body weight were observed; however, the administration of the polyenes mixture reduced food intake	[76]
	Water	Branches	Diabetic rats: 70 mg/mL 90 mg/mL 110 mg/mL Normoglycemic rats: 90 mg/mL	Diabetic and normoglycemic Wistar rats	- Diabetic rats: ↓ Glucose levels with a dose of 70 mg/mL (18.4%) - Normoglycemic rats: ↓ Glucose levels with a dose of 90 mg/mL (33.58%) Insulin levels were not measured		[77]
	DMSO	Cytopiloyne	db/db mice: 0.1 mg/kg 0.5 mg/kg 2.5mg/kg RIN-m5F cells: 7 μM 14 μM 28 μM	db/db mice RIN-m5F (pancreatic β cells)	The doses used are presented: In vivo model: ↓ Postprandial blood glucose levels (61%) ↑ Insulin levels (54%) ↑ Protected pancreatic β-cells. In vitro: ↑ Intracellular insulin concentration with a dose of 28 μM ↑ PKC-α activation with increasing dose ↑ Insulin transcript at 28 µM dose.		[78]
	*Bidens pilosa* formulation (*capsule*)		Humans (n = 10) 400 mg	Diabetic and normoglycemic humans	↓ Fasting glucose levels (39%) ↑ Increasing insulin levels (20%) in normoglycemic subjects		[79]

## Data Availability

No new data were created or analyzed in this study. Data sharing is not applicable to this article.

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
