# Peer review of "Mexican Plants Involved in Glucose Homeostasis and Body Weight Control: Systematic Review"

_nutrients, 2023, doi:10.3390/nu15092070_

Round 1
Reviewer 1 Report
The article is very interesting. However, this requires editorial corrections. There is noticeable chaos at work. I propose to change the layout of the manuscript. First of all, in the title, I suggest entering the species of plants mentioned in the article. Then, in the theoretical part, put photos of these plants and a botanical description and their occurrence in Mexico. This layout will divide the article into specific sections and make it easier to read and study the results
It is worth noting that the authors indicate new research directions.
Comments are listed below:
11. need to improve citation list. The lack of numbering makes it difficult to read the manuscript
22. Line: 848
the given website citation " World Obesity Federation. (2022). World Obesity Atlas 2022 (Issue March). www.worldobesity.org#worldobesityatlas" does not indicate a document. Authors should indicate a more precise citation
33. Line 44. I have doubts about the last sentence of the first paragraph. Of course, I understand that the authors refer to the subject of the article in the sentence, but the sentence suggests that obesity increases the risk of suffering from type 2 diabetes only. I suggest changing the form of the statement.
44. First of all, the seven-page table is a vast collection of information that is extremely difficult to interpret. I suggest that the authors divide the table into 5 different ones, taking into account each plant separately. Each of the tables describing the properties of a different plant should be inserted in the appropriate place in the manuscript.
55. first, there should be a description of individual plant species to introduce the reader to the issue, and then tables with the properties of extracts
66. is the location of the plants proven on the basis of atlases of plant occurrence in Mexico or is it the result of the authors' observations? For example: line: 121
77. line: 126. No citation
88. table 2 is very illegible
99. line 163 – 170: citation is missing
110. line 174: explain the abbreviation p-AMPK
111. the picture description is wrong. point "e" is missing
112. the title of Figure 1 is missing.
113. use abbreviations and symbols in table 1, which will shorten the size of the table and facilitate interpretation
114. in the part devoted to the botanical description, it should be indicated which parts of these plants are edible and which exhibit the properties referred to in the article
Author Response
Thank you for all the suggestions.
Attached you will find the answers to your comments.
I also put them in the website, in case the attached doc does not appear.
- Need to improve citation list. The lack of numbering makes it difficult to read the manuscript.
Thank you for your observation. For the references, we followed the journal´s instructions and the citations must be presented in the Chicago Style (last name and year). We are sorry that we cannot use numbers as you propose, but we must follow the editing instructions.
- Line: 848
The given website citation " World Obesity Federation. (2022). World Obesity Atlas 2022 (Issue March). www.worldobesity.org#worldobesityatlas" does not indicate a document. Authors should indicate a more precise citation.
This citation was changed from World Obesity Federation to:
De Pauw, R., Claessens, M., Gorasso, V., Drieskens, S., Faes, C., & Devleesschauwer, B. (2022). Past, present, and future trends of overweight and obesity in Belgium using Bayesian age-period-cohort models. BMC Public Health, 22(1), 1–14. https://doi.org/10.1186/s12889-022-13685-w.
Specific author´s name and the year. See Line 42 and in the reference list line 873-875.
- Line 44. I have doubts about the last sentence of the first paragraph. Of course, I understand that the authors refer to the subject of the article in the sentence, but the sentence suggests that obesity increases the risk of suffering from type 2 diabetes only. I suggest changing the form of the statement.
We rephrase the sentence as follows: Obesity increases the risk for developing different diseases, such as arterial hypertension, dyslipidemia, metabolic syndrome, type 2 diabetes (T2D), among others (Forouhi & Wareham, 2014).
- First of all, the seven-page table is a vast collection of information that is extremely difficult to interpret. I suggest that the authors divide the table into 5 different ones, taking into account each plant separately. Each of the tables describing the properties of a different plant should be inserted in the appropriate place in the manuscript.
We appreciate the reviewers’ comments, so following his/her advice, we divided the table in 5 different tables for each plant. As suggested, we inserted the tables after each plant´s description.
- first, there should be a description of individual plant species to introduce the reader to the issue, and then tables with the properties of extracts
We restructured the paper as suggested. In the first paragraph of each plant, we mentioned the places where these plants are located/cultivated and the family/genre that they belong to. Then, we described the parts of the plants that edible and that could be used to obtain organic or aqueous extracts. After that, we described the physiological or in vitro properties of those extracts over glucose homeostasis.
- is the location of the plants proven on the basis of atlases of plant occurrence in Mexico or is it the result of the authors' observations? For example: line: 121
The plants´ locations mentioned in this manuscript are based on a Mexican plant atlas and are now cited for each plant.
L130-133
In Mexico, it is grown in Oaxaca, Quintana Roo, Chiapas, Tabasco, Veracruz, and Yucatán (Biblioteca digital de la medicina tradicional mexicana 2009a). It belongs to the Cucurbitaceae family and is colloquially known as melón amargo (Vera & Manzaba, 2019) (Figure 2).
L222-224
Cucurbita ficifolia bouché is a medicinal plant native to America. In Mexico, it is grown in Hidalgo, Guerrero, Michoacán, and Veracruz (Biblioteca digital de la medicina tradicional mexicana 2009b).
L298-300
In Mexico, it is grown in Puebla, Baja California, Chiapas, Oaxaca, Veracruz, Puebla, Hidalgo, Jalisco, Michoacán, Zacatecas, and Sonora (Bioblioteca digital de la medicina tradicional mexicana 2009c).
L368-371
In Mexico, it is grown in Baja California Sur, Sonora, Michoacán, Jalisco, State of Mexico, Hidalgo, Veracruz, Puebla, Tabasco, Oaxaca, Morelos, and Guerrero (Biblioteca digital de la medicina tradicional mexicana 2009d).
L468-471
In Mexico, it is cultivated in Aguascalientes, Durango, Guanajuato, Guerrero, Hidalgo, Jalisco, Michoacán, Morelos, Puebla, Querétaro, Tabasco, Tlaxcala, Veracruz, and Zacatecas (Biblioteca digital de la medicina tradicional mexicana 2009e).
- line: 126. No citation
For that specific sentence, no citation was added, because it is the introduction for all the papers reviewed that are presented in the next section of the manuscript. However, we rephrase the sentence as follows:
- 142-144
In the articles reviewed for this plant, aqueous or organic extracts from leaves, pulp, or seeds were obtained to performed in vivo and in vitro studies to analyze its hypoglycemic effect.
- table 2 is very illegible
We increased the fold of all the text in this table. Also, we inserted it as a table, instead of as an image (as it was before). We think it can now easily be read.
- line 163 – 170: citation is missing
We added the citation (Jiang et al., 2020). It reads as follows:
L181-184
The hypoglycemic effect of saponins from the hydroalcoholic extract of the pulp of Momordica charantia l. was evaluated. The saponins were administered for 4 weeks, using doses of 400, 200 and 100 mg/kg (Table 1). This study showed that saponins can reduce glucose levels (10%) and insulin resistance in diabetic rats (Jiang et al., 2020).
- line 174: explain the abbreviation p-AMPK
Now it says:
L192-193
In addition, the extracts increased p-AMPK (phosphorylated AMP-activated protein kinase) and improved antioxidant capacity, exhibiting a protective effect on pancreatic β cells.
- the picture description is wrong. point "e" is missing
In this new version of the manuscript, each plant picture is individually presented, so no need for “e”.
- the title of Figure 1 is missing.
It has been added to the new version of the manuscript. L100
- use abbreviations and symbols in table 1, which will shorten the size of the table and facilitate interpretation
Table 1 was divided in 5 different tables showing one plant per table. Moreover, arrows added to make it more didactic and easier to read.
- in the part devoted to the botanical description, it should be indicated which parts of these plants are edible and which exhibit the properties referred to in the article
We appreciate your suggestion. We added a paragraph describing the edible parts, after the image of each plant.
L.140-141 Momordica charantia l. produces an edible fruit which is harvested for cooking. The seeds and skin are also edible (Vera & Manzaba, 2019).
L.241-242 The skin, pulp, and seeds are edible. In Mexico, the pulp and seeds are widely used to prepare different dishes and regional sweets (Andrade-Cetto & Heinrich, 2005).
L.308-309 The entire plant is edible (leaves and seeds), it is commonly used as a condiment in different dishes (Rodríguez, 2015).
L.378-379 The most common edible part of this plant is the pulp; however, the leaves are used as a spice (Angel., 2003).
L.479-480 This plant it is widely used as fodder for domestic animals. It also has an edible flower that is used in salads; the shoots and leaves are used as potherbs (Yang, 2014).
Reviewer 2 Report
Comments:
Table 1. Summary of reviewed articles of 5 selected Mexican medicinal plants. Most of the experimental models used are animal studies. Only one or two human studies (Lai st al., 2015; and Ahmed et al., 2019) were included. Have you found more human trials using Mexican medicinal plants (not limited to these 5 selected Mexican plants). For Ahmed et al., 2019, it was mixed with a mouse study, it was not clear the Pulp (Avocatin B) had any effects on Body weight or glucose levels or insulin secretion in humans.
Line 179, (and in Table 1, by Chang et al., 2015) FL83B is a hepatocyte cell that was isolated from the liver of a 15 to 17 day old fetal mouse. So you should indicate FL83B is a mouse cell line not human.
Section 4. Discussion, lines 422-423. “Mexican plants have been used for many centuries for the treatment of T2D and recently for body weight control”. More details (which countries have approved to use these plants? incl. guidelines) should be given with References.
Are these any studies in comparison of any of these 5 medicinal Mexican plants with metformin?
Author Response
Comments:
- Table 1. Summary of reviewed articles of 5 selected Mexican medicinal plants. Most of the experimental models used are animal studies. Only one or two human studies (Lai st al., 2015; and Ahmed et al., 2019) were included. Have you found more human trials using Mexican medicinal plants (not limited to these 5 selected Mexican plants).
L64-70
Medicinal plants have been used empirically (either as part of the diet, infusions, or extracts) to treat and improve T2D symptoms and body weight control. Despite being used empirically, the World Health Organization has guidelines on the safety monitoring of herbal medicines (World Health Organization (WHO) 2004). Different medicinal plants are now approved and recommended for its medicinal use after they have been scientifically validated to ensure safety and efficacy, like, Echinacea purpurea, Panax ginseg, Passiflora Incarnata, and others. However, the use of medicinal plants is not yet approved by the FDA (Principe and Jose 2002).
- For Ahmed et al., 2019, it was mixed with a mouse study, it was not clear the Pulp (Avocatin B) had any effects on Body weight or glucose levels or insulin secretion in humans.
- 441-449
On the other hand, a pilot study was carried out with humans (n=10), administering them 50 mg or 200 mg per day for 60 days of avocatin B, observing that its consumption had no effect on blood markers of kidney, liver, and muscle toxicity (total bilirubin, alanine aminotransferase, creatinine, and creatine phosphokinase), concluding that the supplementation with avocatin B does not generate discomfort or harmful effects. Insulin secretion and glucose levels were not measured in humans, because the objective was to assert the avocatin B toxicity in humans. However, it was reported that 200 mg of avocatin B per day reduced body weight by 6 % (Ahmed et al. 2019).
- Line 179, (and in Table 1, by Chang et al., 2015) FL83B is a hepatocyte cell that was isolated from the liver of a 15 to 17 day old fetal mouse. So you should indicate FL83B is a mouse cell line not human.
We appreciate your observation. We added mouse cell line as suggested. It now says:
L.197-200
Other compounds found in Momordica charantia l. extracts are the triterpenes that used at 20 and 30 μM in a mouse cell line of hepatocytes (FL83B), increased glucose uptake and insulin receptor 1 (IRS-1) phosphorylation in insulin resistant cells, which could indicate the improvement of insulin sensitivity.
- Section 4. Discussion, lines 422-423. “Mexican plants have been used for many centuries for the treatment of T2D and recently for body weight control”. More details (which countries have approved to use these plants? incl. guidelines) should be given with References.
We thank the reviewer for his comment and, as suggested, added a paragraph to enrich our manuscript.
L.544-.557
Mexican plants have been used for many centuries for the treatment of T2D and recently for body weight control. In México, the Federal Commission for the Protection against Sanitary Risk (COFEPRIS) allowed the use of 18 medicinal plants as herbal medicine (Comisión Federal para la Protección contra los Riesgos Sanitarios 2018). For example, Echinacea purpurea is approved in Mexico, Belgium, Bulgaria, Croatia, and Denmark with a dose of 100 mg (Committee on Herbal Medicinal Products (HMPC) 2014). Another example is Panax ginseg, which is approved in Mexico, Austria, Belgium, Denmark, France, Germany, Ireland, Portugal, and Spain with a dose of 100 mg or 300 mg (Committee on Herbal Medicinal Products 2014). Moreover, Passiflora Incarnata is also approved in Mexico, Germany, Austria, and Belgium with a dose of 200 mg (Committee on Herbal Medicinal Products (HMPC) 2014). For herbal compounds commercialization, the advertising and marketing must include therapeutic indication and symptomatic effect, in no case these products could be advertised as curatives (Rodríguez-Hernández et al. 2022).
- Are these any studies in comparison of any of these 5 medicinal Mexican plants with metformin?
We added some studies were the plants´extracts have been compared to metformin. However, we didn’t find these studies for all the plants.
L204.210
In the articles reviewed of this plant, the different extracts were compared primarily with glibenclamide (antidiabetic drug) having a similar hypoglycemic effect (50%) and restoring body weight loss (Ofuegbe et al. 2019). In addition, studies comparing the effect of these extracts against metformin (antidiabetic drug) demonstrated reduction of blood glucose levels of 50.1% using 300 mg/kg plant extract and 47% for metformin (Islam et al. 2018; Pramesthi, Ardana, and Indriyanti 2019; Poonam, Prem Prakash, and Vijay Kumar 2013)
L286-288
In the articles reviewed for this plant, the different extracts have been compared primarily with tolbutamide (83% reduction in blood glucose levels) (Alarcon-Aguilar et al. 2002) and not with metformin (antidiabetic drugs).
L 350-355
The effects of Coriandrum sativum l. extract was evaluated comparing to glibenclamide, demonstrating a similar hypoglycemic effect (20%) and improvement in insulin resistance (Aissaoui et al. 2011). Moreover, in another study, blood glucose levels were evaluated comparing the effects of the plant (49 % reduction with dose of 40mg/kg) with metformin (61% reduction), but neither insulin secretion or body weight were evaluated in this study (Das et al. 2019).
L450-455
In the articles reviewed of this plant, the different extracts were compared primarily with glibenclamide having a similar hypoglycemic effect (58.9%), restoring body weight loss and a protective effect in the pancreas (Ezejiofor, Okorie, and Orisakwe 2013). On the other hand, there have been studies where the hypoglycemic effects of the plant (reduction of 79% with a dose of 53.3 mg/kg) have been compared with metformin (reduction of 80%) (Ojo et al. 2022).
L526-528
In the articles reviewed for this plant, the different extracts were compared primarily with glibenclamide, having a similar hypoglycemic effect (50%) (Hsu et al. 2009). On the other hand, there have not been studies where it has been compared with metformin.